# Disentangling Identity Features from Interference Factors for Cloth-Changing Person Re-Identification

## ABSTRACT

Cloth-Changing Person Re-Identification (CC-ReID) aims to accurately identify a target person in the more realistic surveillance scenario where clothes of the pedestrian may change drastically, which is critical in public security systems for tracking down disguised criminal suspects. Existing methods mainly transform the CC-ReID problem into cross-modality feature alignment from the data-driven perspective, without modelling the interference factors such as clothes and camera view changes meticulously. This may lead to over-consideration or under-consideration of the influence of these factors on the extraction of robust and discriminative identity features. This paper proposes a novel algorithm for thoroughly disentangling identity features from interference factors brought by clothes and camera view changes while ensuring the robustness and discriminability. It adopts a dual-stream identity feature learning framework consisting of a raw image stream and a cloth-erasing stream, to explore discriminative and cloth-irrelevant identity feature representations. Specifically, an adaptive cloth-irrelevant contrastive objective is introduced to contrast features extracted by the two streams, aiming to suppress the fluctuation caused by clothes textures in the identity feature space. Moreover, we innovatively mitigate the influence of the interference factors through a generative adversarial interference factor decoupling network. This network is targeted at capturing identity-related information residing in the interference factors and disentangling the identity features from such information. Extensive experimental results demonstrate the effectiveness of the proposed method, achieving superior performances to state-of-the-art methods. Our source code is available in the supplementary materials.

## CCS CONCEPTS

• **Computing methodologies** → **Object identification**.

## KEYWORDS

Cloth-Changing Person ReID, Contrastive Learning, General Adversarial Learning

## 1 INTRODUCTION

Person re-identification (ReID) aims to search for a person of interest across non-overlapping cameras, which has very important application values in intelligent surveillance system. Existing general

*ACM Multimedia, June 03–05, 2024, Woodstock, NY*
© 2024 Copyright held by the owner/author(s). Publication rights licensed to ACM.
ACM ISBN 978-1-4503-XXXX-X/24/06
https://doi.org/XXX.XXX

**Unpublished working draft. Not for distribution.**

ReID methods primarily focus on identifying the same pedestrian only on the short-term scenarios with limited time and space span, under the assumption that each individual maintains consistent clothes and accessories. On the contrary, the cloth-changing person re-identification (CC-ReID) concentrates on the more challenging and realistic long-term surveillance scenario where the clothes of the pedestrian may change drastically, which is critical in public security systems for tracking down disguised criminal suspects. Obviously, apart from the general challenges, such as moderate view/pose changes, occlusion, low resolution etc., the CC-ReID faces more severer challenges, *i.e.*, clothes changes and drastic view changes caused by the long-term applications.

Considering that the pedestrian clothes and camera views are often coupled with the identity information, they interfere with the learning of robust identity features. Disentangling identity features from those interference factors is the key to address CC-ReID. Such disentanglement is not a trivial problem. For example, roughly erasing all information from the clothing areas in the image space not only destroys the integrity of the features but also causes the model to lose critical information from the clothing areas.

Existing CC-ReID methods mainly focus on learning identity feature representations which are relevant to stable identity-related factors rather than the interference factors, like cloth and environmental characteristics. They can be roughly divided into single-modality and multi-modality based methods. Specifically, the single-modality methods [9, 12, 18, 40] learn cloth-irrelevant identity features solely from the RGB images, which are insufficient for removing the influence of interference factors. The multi-modality methods improve the performance by exploring other synthesized data modalities such as shapes [16], gait motions [21], cloth-erased images [31], or cloth-exchanged images [10] to capture cloth-irrelevant identity features. They mainly transform the interference factor disentanglement problem into cross-modality feature alignment from the data-driven perspective. The auxiliary data modalities used in [16, 21] only preserve limited identity-related information, leading to the under-exploration of identity features. Those methods [10, 31] using cloth-erased images or cloth-exchanged images lack intrinsic consideration of the influence of clothes characteristics and neglect the influence of other underlying interference factors. These issues make them very hard to ensure the robustness of the learned identity features.

In this paper, we advocate for disentangling identity features from interference factors in a novel reverse perspective. Namely, we propose to enhance the robustness of identity features through suppressing the intra-pedestrian variances caused by clothes textures and relieving the dependence on interference factors during the discriminative learning of identity features. First, we propose a dual-stream identity feature learning framework, consisting of a raw image stream and a cloth-erasing stream. The cloth-erasing stream is implemented based on the image parsing method SCHP [24], achieving the extraction of identity features unrelated to clothes

textures. This dual-stream framework aims to suppress the fluctuations caused by diversified cloth textures in the identity feature space, hence deriving cloth-irrelevant identity feature representations. A cloth-irrelevant contrastive learning objective is introduced to constrain the distance between features extracted by the raw image and cloth-erasing streams within the minimal intra-pedestrian distance. It contributes to building more compact intra-pedestrian relationship structures among the identity features of pedestrian images with diverse clothes. Such intra-pedestrian relationship structures greatly help to obtain cloth-irrelevant identity features, which could mitigate the variance caused by clothes textures. Furthermore, to reduce the negative effects caused by the inaccurate image parsing result for cloth erasure, we further propose to adaptively reweight the cloth-irrelevant contrastive learning objective according to the discriminability of the cloth-erased pedestrian image.

To mitigate the influence of interference factors more thoroughly, we also propose a generative adversarial interference factor decoupling network. Specifically, we first design an interference factor recognition branch to capture the interference-factor-oriented features. Simultaneously, we design a feature disentanglement module for extracting identity-related information from the interference-factor-oriented features. This module is learned in the cycle-consistent adversarial manner [44], guaranteeing the homogeneity between the generated features and the identity features. Finally, we make the discriminative learning process of identity features independent of identity-related information.

In summary, our main contributions are as follows:

- We propose a dual-stream identity feature learning framework to obtain discriminative cloth-irrelevant identity feature representations. Specifically, an adaptive cloth-irrelevant contrastive learning objective is introduced to suppress the fluctuations caused by clothes textures in the identity feature space.
- To further enhance the robustness of identity features, we introduce a generative adversarial interference factor decoupling network, which can relieve the dependence on interference-factor-related identity information during the discriminative learning of identity features.
- Extensive experimental results greatly demonstrate the effectiveness of the proposed method in learning discriminative identity features for CC-ReID, which achieves superior performance to state-of-the-art methods.

## 2 RELATED WORK

### 2.1 Person Re-Identification

Person ReID aims to retrieve images of the same individual from a gallery taken by non-overlapping cameras in the RGB image space. Existing deep learning-based person ReID methods generally fall into two main categories. Firstly, **improvement of pedestrian feature extraction network structures** [32, 36, 42], involves extracting fine-grained local features to enhance the model's attention to detail features. For instance, [32] extracts local features by dividing the feature map into vertical parts. Secondly, **development of metric learning loss functions** [3, 11, 35], involves introducing contrastive loss [35] and triplet loss [11] into person ReID.

However, these traditional person ReID methods have encountered significant performance degradation under poor lighting conditions. Consequently, many researchers have proposed valuable methods in cross-modality person ReID.

Unlike single-modality person ReID methods, the main challenge of cross-modality person ReID lies in the heterogeneous modality discrepancy [4], with most methods achieving this by learning modality-shared information to suppress modality discrepancy. Some approaches [6, 23, 37] achieve image-level modality alignment by generating intermediate modalities, while others achieve feature-level modality alignment through metric learning methods [13, 41]. Cross-modality person ReID and Cloth-changing person ReID(CC-ReID) share similarities in that, both require exploring shared knowledge and suppressing the drastic variances within intra-pedestrian images. Unlike person ReID with RGB images, where clothing remains constant, CC-ReID presents a greater challenge by requiring the model to extract identity features that are not influenced by clothing changes.

### 2.2 Cloth-changing person ReID

CC-ReID becomes increasingly critical for long-term person ReID in surveillance scenarios. To address the challenge of changing clothes over time, an increasing number of researchers focus on CC-ReID and have made significant contributions. Existing CC-ReID methods can be roughly categorized into single-modality and multi-modality methods.

Single-modality methods [9, 12, 18, 40] usually aim to explore cloth-irrelevant features just from RGB images. For instance, [9] penalizes the model's ability to predict clothes by an additional clothing classifier, [40] proposes to reduce the information from clothing through causal intervention. Although single-modality methods reduce the workload during data preparation, the scarcity of clothes diversity in training images makes it difficult to achieve better results from a single modality alone. Multi-modality works [10, 26, 31] adopt human parsing maps to obtain the foreground cloth region, creating intermediate images through cloth erasure. Besides, gait information [21] and shape information [16] are also adopted to make the model focus more on the pedestrian's latent identity information. Some methods [20, 38] also utilize GANs to change clothes of images for enriching the dataset. These multi-modality methods often rely too much on the quality of the generated images. When the quality of the generated images is low, the model may learn irrelevant features. Unlike these methods, we first design a Dual-stream Identity Feature Learning Framework (DSIFLF) to encourage the model to ignore the texture information of clothing. Furthermore, a specifically designed generative adversarial interference factor decoupling network (GAIFDN) is utilized to eliminate interference information, *i.e.*, clothes and viewpoint changes, thus enhancing the robustness of identity features.

### 2.3 Generative Adversarial Networks

Generative Adversarial Networks (GANs) involve continuous adversarial training between a generator and a discriminator, ultimately producing data that the discriminator finds challenging to differentiate. GANs are originally proposed from the perspective of game

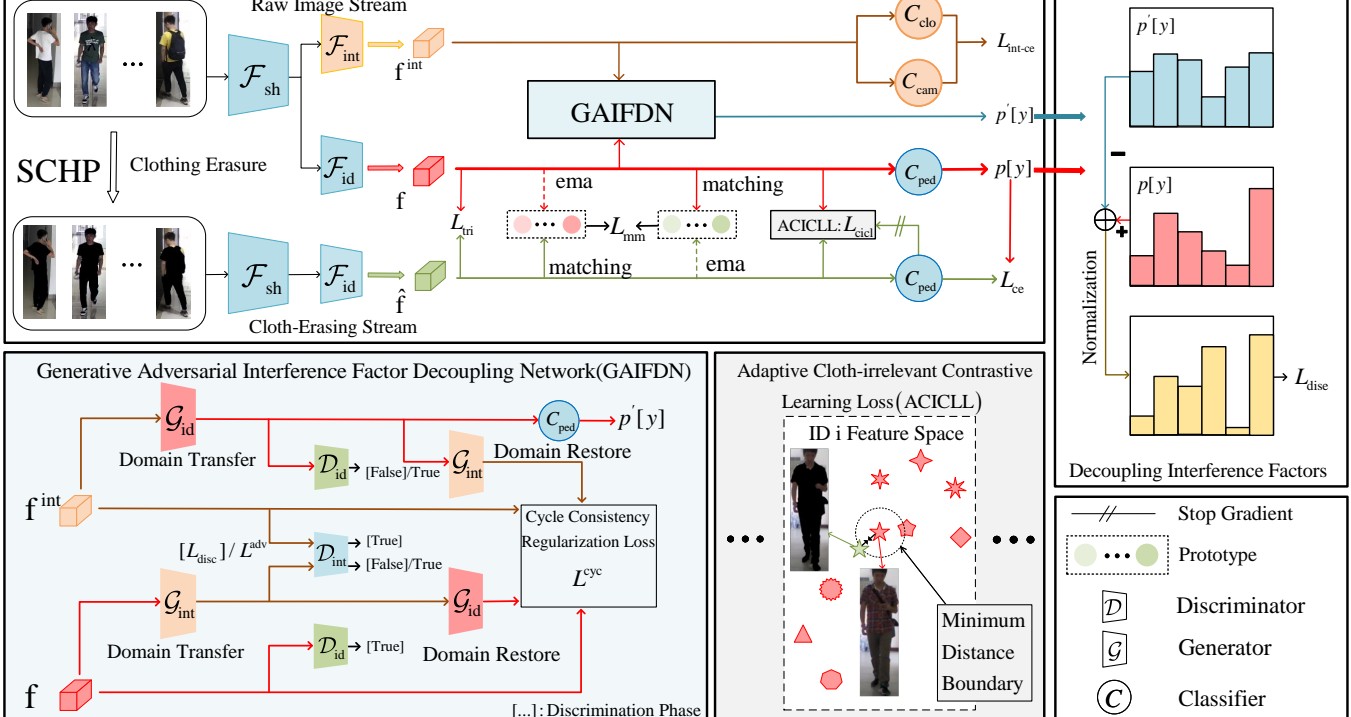

**Figure 1: The overall framework of our method, which includes two important components of our model: *Dual-stream Identity Feature Learning Framework* (DSIFLF) and *Generative Adversarial Interference Factor Decoupling Network* (GAIFDN). DSIFLF consists of raw image stream and cloth-erasing stream, where 'ema' represents the exponential moving average update of prototypes.**

theory [8], and thereafter, they are widely applied in the field of image generation [1, 19, 28, 44]. Among these works, CycleGAN [44] achieves image generation through cycle consistency constraints. GANs have many applications in CC-ReID [12, 20, 38]. However, unlike these methods that utilize GANs to change the clothes in pedestrian images, we design GAIFDN, which extracts identity-related information from the interference-factor-oriented features. Through disentanglement methods, the model is able to learn more robust and discriminative identity features.

## 3 METHOD

### 3.1 Problem Definition

Let $\mathbb{X} = \{(\mathbf{X}_i, y_i, y_i^{\text{clo}}, y_i^{\text{cam}})\}_{i=1}^N$ represents the training dataset containing $N$ images from $M_{\text{ped}}$ pedestrians captured with $M_{\text{cam}}$ cameras. In practical scenarios, the same pedestrian may appear wearing different clothes, and we assume the total number of clothes in $\mathbb{X}$ is $M_{\text{clo}}$. Here, $\mathbf{X}_i \in \mathbb{R}^{H \times W \times C}$ denotes the $i$-th pedestrian image, where $H$, $W$, and $C$ represent the height, width, and number of channels, respectively. $y_i \in [1, M_{\text{ped}}]$, $y_i^{\text{clo}} \in [1, M_{\text{clo}}]$, and $y_i^{\text{cam}} \in [1, M_{\text{cam}}]$ denote the pedestrian identity, clothes number, and camera number of $\mathbf{X}_i$, respectively. The target of this paper is to learn a feature extractor capable of discriminating whether two images belong to the same pedestrian ignoring clothes and camera differences.

### 3.2 Overview

As illustrated in Fig. 1, we use ResNet50 [14] pre-trained on ImageNet [5] as the backbone for extracting discriminative identity features from pedestrian images which are robust against interference factors including clothing and camera view changes. Based on the backbone, we first set up a Dual-stream Identity Feature Learning Framework (DSIFLF) which is constituted by a raw image stream and a cloth-erasing stream, for learning identity features which are independent of the clothing textures. To achieve this target, we introduce a novel adaptive cloth-irrelevant contrastive learning loss constraining the outputs of the two streams. Moreover, we devise a Generative Adversarial Interference Factor Decoupling Network to more thoroughly mitigate the influence of clothing and camera view changes on the robustness of identity features. Concretely, an interference factor recognition branch is first utilized to capture the information related to clothing and camera specifications. Then, we leverage the cyclic generative adversarial learning algorithm [44] to foster a feature disentanglement module which aims to squeeze identity-related information from the features captured by the interference factor recognition branch. Finally, we design an interference factor decoupling objective to guide the learning of identity features without depending on the information related to clothing and camera specifications. Technical details are introduced in the following subsections.

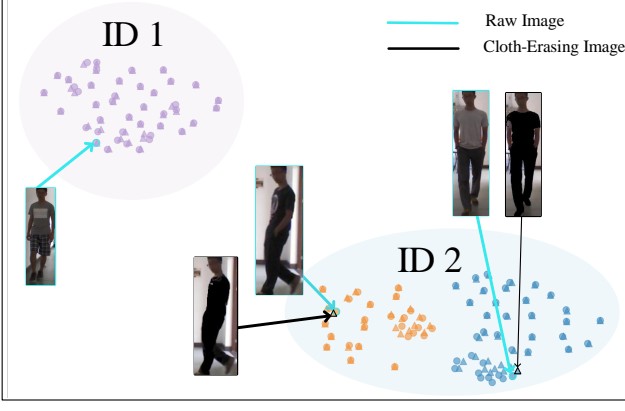

**Figure 2: For the t-SNE [33] visualization of two pedestrians, within each ID, different colors represent pedestrians wearing different clothes. "○" represents the features of our original images, while "△" represents the features of images with clothes erased. It can be observed that, given an image in which the pedestrian wearing any clothes, the feature distance between it and its cloth-erased counterpart is usually to be smaller than the intra-pedestrian feature distance.**

## 3.3 Dual-stream Identity Feature Learning Framework (DSIFLF)

The change of clothes increases the intra-pedestrian appearance variance due to the texture differences between different clothes, which increases the variance in the identity feature space as illustrated by Fig. 2. This increases the difficulty in the ReID of pedestrians wearing substantially different clothes and the discrimination between different pedestrians wearing similar clothes. To address this issue, we propose a novel dual-stream identity feature learning framework through binding a raw image and a cloth-erasing streams. Conventional pedestrian-discriminative feature learning, prototype-based cross-stream matching, and novel cloth-invariant contrastive learning objectives are adopted for optimization in this framework.

*3.3.1 Dual-stream Identity Feature Extraction.* We split the ResNet50 backbone into two components: a shared image encoder (denoted as $\mathcal{F}_{\text{sh}}(\cdot)$) composed of the first four layer blocks, and an identity feature extractor (denoted as $\mathcal{F}_{\text{id}}(\cdot)$) consisting of the fifth layer block. A pedestrian classifier denoted as $C_{\text{ped}}(\cdot)$ is attached after $\mathcal{F}_{\text{id}}(\cdot)$ to recognize the pedestrian identification of input images. The calculation process of the raw image stream is as follows. Given an input image $\mathbf{X}_i$, a $D$-dimensional identity feature vector denoted as $\mathbf{f}_i$ can be generated via $\mathbf{f}_i = \mathcal{F}_{\text{id}}(\mathcal{F}_{\text{sh}}(\mathbf{X}_i))$. The pedestrian classifier predicts a $M_{\text{ped}}$-dimensional probability vector denoted as $\mathbf{p}_i$ from $\mathbf{f}_i$, i.e., $\mathbf{p}_i = C_{\text{ped}}(\mathbf{f}_i)$. The cloth-erasing stream is implemented as follows. First, we extract the clothes region using the SCHP model [24], which derives a binary mask $\mathbf{S}_i$. Then, an augmented counterpart (denoted as $\hat{\mathbf{X}}_i$) for $\mathbf{X}_i$ is obtained via $\hat{\mathbf{X}}_i = \mathbf{X}_i \circ (1 - \mathbf{S}_i)$, where ○ denotes the element-wise product. Such a clothes masking operation is capable of removing the clothes textures. We feed $\hat{\mathbf{X}}_i$ through the shared image encoder and identity feature extractor,

which derives the other feature vector $\hat{\mathbf{f}}_i$, i.e., $\hat{\mathbf{f}}_i = \mathcal{F}_{\text{id}}(\mathcal{F}_{\text{sh}}(\hat{\mathbf{X}}_i))$. Finally, we input $\hat{\mathbf{f}}_i$ into the pedestrian classifier, generating a probability vector $\hat{\mathbf{p}}_i$, i.e., $\hat{\mathbf{p}}_i = C_{\text{ped}}(\hat{\mathbf{f}}_i)$.

*3.3.2 Pedestrian-Discriminative Feature Learning.* The cross-entropy loss function is applied for constraining the predictions $\mathbf{p}_i$ and $\hat{\mathbf{p}}_i$ of the pedestrian classifier:

$$L_{\text{ce}} = -\frac{1}{N}\sum_{i=1}^{N}\{\log(p_i[y_i]) + \log(\hat{p}_i[y_i])\}, \tag{1}$$

where $p_i[m]$ and $\hat{p}_i[m]$ denote the $m$-th elements of $\mathbf{p}_i$ and $\hat{\mathbf{p}}_i$, respectively. The above objective can make the backbone model focus on extracting features capable of discriminating pedestrian identities.

Moreover, the triplet loss [15] denoted as $L_{\text{tri}}$ is used to regularize the outputs of the feature extractor for improving the inter-pedestrian discriminability:

$$L_{\text{tri}} = \frac{1}{N}\sum_{i=1}^{N}\sum_{j\in\mathbb{S}_i^+, k\in\mathbb{S}_i^-}[\max(\text{dist}(\mathbf{f}_i, \mathbf{f}_j) - \text{dist}(\mathbf{f}_i, \mathbf{f}_k) + \epsilon, 0) \\ + \max(\text{dist}(\hat{\mathbf{f}}_i, \hat{\mathbf{f}}_j) - \text{dist}(\hat{\mathbf{f}}_i, \hat{\mathbf{f}}_k) + \epsilon, 0)], \tag{2}$$

where $\mathbb{S}_i^+$ represents the indices of images having the same pedestrian identity with $\mathbf{X}_i$ and $\mathbb{S}_i^-$ represents those of images having different pedestrian identities with $\mathbf{X}_i$. $\text{dist}(\mathbf{f}_i, \mathbf{f}_j)$ calculates the cosine distance between $\mathbf{f}_i$ and $\mathbf{f}_j$, i.e., $\text{dist}(\mathbf{f}_i, \mathbf{f}_j) = 1 - \cos(\mathbf{f}_i, \mathbf{f}_j)$. $\epsilon$ is a constant.

*3.3.3 Prototype-based Cross-stream Matching.* The cloth-erased images have a distinct feature distribution compared to raw images. This impacts the representation learning of the shared image encoder and the identity feature extractor. To address this issue, we devise a prototype-based cross-stream matching objective for pulling close the feature distributions of raw and cloth-erased images. First, for estimating the feature distributions of the $m$-th pedestrian, we construct two prototypes denoted as $\mathbf{o}_m$ and $\hat{\mathbf{o}}_m$ by accumulating features of raw and cloth-erased images, respectively. The exponential moving average is used to achieve the calculation of $\mathbf{o}_m$ and $\hat{\mathbf{o}}_m$, i.e.,

$$\mathbf{o}_m \leftarrow \gamma \times \mathbf{o}_m + (1 - \gamma) \times \frac{\sum_{i=1}^{N}\mathbb{1}(y_i = m)\mathbf{f}_i}{\sum_{i=1}^{N}\mathbb{1}(y_i = m)}, \tag{3}$$

$$\hat{\mathbf{o}}_m \leftarrow \gamma \times \hat{\mathbf{o}}_m + (1 - \gamma) \times \frac{\sum_{i=1}^{N}\mathbb{1}(y_i = m)\hat{\mathbf{f}}_i}{\sum_{i=1}^{N}\mathbb{1}(y_i = m)}, \tag{4}$$

where $\mathbb{1}(\cdot)$ is the indicator function. The second term of Eq. (3) or (4) aims to calculate the average of each pedestrian's identity features extracted by the current model. The averaged identity features are used to update the pedestrian prototypes iteratively. $\gamma$ is a hyper-parameter controlling the updating rate of pedestrian prototypes.

Afterwards, we calculate the prototype-based cross-stream matching loss as follows,

$$L_{\text{mm}} = -\frac{1}{N} \sum_{i=1}^{N} \left[ \log\left(\frac{\exp(\mathbf{f}_i \cdot \hat{\mathbf{o}}_{y_i})}{\sum_{m=1}^{M_{\text{ped}}} \exp(\mathbf{f}_i \cdot \hat{\mathbf{o}}_m)}\right) \right.$$
$$\left. + \log\left(\frac{\exp(\hat{\mathbf{f}}_i \cdot \mathbf{o}_{y_i})}{\sum_{m=1}^{M_{\text{ped}}} \exp(\hat{\mathbf{f}}_i \cdot \mathbf{o}_m)}\right) \right]. \tag{5}$$

'·' denotes the inner product operation. The above loss attracts the identity features of one stream to the prototypes estimated from identity features of the other stream, hence achieving the target of reducing the feature distribution gap between raw and cloth-erased images.

*3.3.4 Adaptive Cloth-irrelevant Contrastive Learning Loss.* For mitigating the variance caused by clothes textures, we propose a adaptive cloth-irrelevant contrastive learning loss objective which helps foster identity features with more compact intra-pedestrian relationship structures. First, we estimate the minimal intra-pedestrian distance of each raw image in the identity feature space. Subsequently, we use it to restrict the distance between identity features of the raw and cloth-erased images as follows,

$$L_{\text{cicl}} = \frac{1}{N} \sum_{i=1}^{N} \text{sg}(\omega_i) \times \max\left\{ \text{dist}(\mathbf{f}_i, \hat{\mathbf{f}}_i) - \min_{j \in \mathbb{S}_i^+} \text{dist}(\mathbf{f}_i, \mathbf{f}_j) + \epsilon, 0 \right\}. \tag{6}$$

$\text{sg}(\cdot)$ denotes the stop-gradient operation. $\omega_i$ is a weighting coefficient for measuring the identity discriminability of $\hat{\mathbf{X}}_i$, which is calculated from the prediction of the pedestrian classifier namely $\hat{\mathbf{p}}_i$. The calculation formulation is as below,

$$\omega_i = \frac{\exp(\hat{p}_i[y_i]/\tau)}{\sum_{m=1}^{M_{\text{ped}}} \exp(\hat{p}_i[m]/\tau)}, \tag{7}$$

where $\tau$ is a temperature coefficient. The target of the loss term in Eq. (6) is: given an image in which the pedestrian wearing any clothes, the feature distance between it and its cloth-erased counterpart is constrained to be smaller than the minimal intra-pedestrian feature distance. This means that the differences to the identity features brought by the cloth textures would not surpass the normal feature fluctuation caused by other factors such as pose changes, under the condition of the same pedestrian identification. Hence, this loss term is beneficial for fostering identity features which are irrelevant to the change of clothes textures. The introduction of the weighting coefficient is motivated by the phenomenon that, the segmented clothes regions may be inaccurate, leading to severe corruption on the identity discriminability for cloth-erased images. The importance of such images needs to be weakened in Eq. (6).

The overall objective denoted as $L_{\text{dual}}$ for the dual-stream identity feature learning framework is formed by combining Eq. (1), (2), (5), and (6):

$$L_{\text{dual}} = L_{\text{ce}} + L_{\text{tri}} + \beta(L_{\text{mm}} + L_{\text{cicl}}), \tag{8}$$

where $\beta$ is a constant. This framework ensures the learned identity features to be discriminative among different pedestrians while irrelevant to diversified clothes textures. In the subsequent subsection, we concentrate on decoupling identity features from interference factors more thoroughly.

## 3.4 Generative Adversarial Interference Factor Decoupling Network (GAIFDN)

The pedestrian identification is moderately coupled with clothes and camera specifications, e.g., one person may frequently wear the same clothes or appear under the same camera view in a period of time. The pedestrian classifier may inadvertently rely on such factors for accomplishing the task of pedestrian recognition, which hinders the exploration of robust identity features. To effectively counteract the influence of interference factors including clothes and camera view changes on identity features, we propose a generative adversarial interference factor decoupling network which is composed of an interference factor recognition branch, a generative adversarial feature disentanglement module, and an interference factor decoupling objective.

*3.4.1 Interference Factor Recognition Branch.* We position a specific feature extractor denoted as $\mathcal{F}_{\text{int}}(\cdot)$ after the shared image encoder. This configuration facilitates the extraction of features specifically targeted at recognizing interference factors. We denote the features extracted for $\mathbf{X}_i$ as $\mathbf{f}_i^{\text{int}}$, i.e., $\mathbf{f}_i^{\text{int}} = \mathcal{F}_{\text{int}}(\mathcal{F}_{\text{sh}}(\mathbf{X}_i))$. Subsequently, we employ two specialized classifiers to ascertain the clothing and camera numbers from $\mathbf{f}_i^{\text{int}}$, yielding outputs $\mathbf{p}_i^{\text{clo}}$ and $\mathbf{p}_i^{\text{cam}}$, respectively.

The training of these classifiers is governed by a cross-entropy loss function, designed to fine-tune their accuracy in predicting clothing and camera numbers:

$$L_{\text{int-ce}} = -\frac{1}{N} \sum_{i=1}^{N} \{\log(p_i^{\text{clo}}[y_i^{\text{clo}}]) + \log(p_i^{\text{cam}}[y_i^{\text{cam}}])\}. \tag{9}$$

*3.4.2 Generative Adversarial Feature Disentanglement Module.* To squeeze identity-related information from the above interference-factor-oriented features, we introduce a generative adversarial feature disentanglement module, inspired from [44]. This involves the construction of a feature generator $\mathcal{G}_{\text{id}}(\cdot)$, which extracts out identity-related information from the interference-factor-oriented features; the other feature generator $\mathcal{G}_{\text{int}}(\cdot)$ is targeted at grasping interference-factor-related features from identity features. Two discriminators, denoted as $\mathcal{D}_{\text{id}}(\cdot)$ and $\mathcal{D}_{\text{int}}(\cdot)$, are used to assess the authenticity of the features within the domains of identity features and interference-factor-oriented feature, respectively.

The discriminators are optimized through an objective function that enhances their ability to discern real identity features from those translated from interference-factor-oriented features, and conversely:

$$L_{\text{disc}} = -\frac{1}{N} \sum_{i=1}^{N} [\log(\mathcal{D}_{\text{id}}(\mathbf{f}_i)) + \log(1 - \mathcal{D}_{\text{id}}(\mathcal{G}_{\text{id}}(\mathbf{f}_i^{\text{int}})))$$
$$+ \log(\mathcal{D}_{\text{int}}(\mathbf{f}_i^{\text{int}})) + \log(1 - \mathcal{D}_{\text{int}}(\mathcal{G}_{\text{int}}(\mathbf{f}_i)))]. \tag{10}$$

In the above objective function, the first two terms constrain the learning of $\mathcal{D}_{\text{id}}(\cdot)$, while the last two terms constrain the learning of $\mathcal{D}_{\text{int}}(\cdot)$.

The generators are subject to an objective ($L_{\text{gen}}$) comprising an adversarial loss ($L_i^{\text{adv}}$) and a cycle consistency regularization $L_i^{\text{cyc}}$:

$$L_{\text{gen}} = \frac{1}{N} \sum_{i=1}^{N} (L_i^{\text{cyc}} + L_i^{\text{adv}}), \tag{11}$$

$$L_i^{\text{adv}} = \log(1 - \mathcal{D}_{\text{id}}(\mathcal{G}_{\text{id}}(\mathbf{f}_i^{\text{int}}))) + \log(1 - \mathcal{D}_{\text{int}}(\mathcal{G}_{\text{int}}(\mathbf{f}_i)))], \tag{12}$$

$$L_i^{\text{cyc}} = ||\mathcal{G}_{\text{int}}(\mathcal{G}_{\text{id}}(\mathbf{f}_i^{\text{int}})) - \mathbf{f}_i^{\text{int}}||_2^2 + ||\mathcal{G}_{\text{id}}(\mathcal{G}_{\text{int}}(\mathbf{f}_i^{\text{id}})) - \mathbf{f}_i^{\text{id}}||_2^2. \tag{13}$$

The adversarial loss in Eq. (12) effectively facilitates feature domain translation, while the consistency regularization in Eq. (13) maintains a cohesive relationship between identity and interference-factor-oriented features.

*3.4.3 Interference Factor Decoupling Objective.* To remove the contribution of interference factors, we reallocate the output probabilities of the pedestrian classifier. Then, the following objective is used to constrain the reallocated output probabilities:

$$L_{\text{dise}} = -\frac{1}{N} \sum_{i=1}^{N} \log(p_i[y_i] - p_i'[y_i]). \tag{14}$$

where $\mathbf{p}_i' = C_{\text{ped}}(\mathcal{G}_{\text{id}}(\mathbf{f}_i^{\text{int}}))$. This loss function ensures the classification decision-making process becomes independent of interference factors, thereby directing the feature extractor to disregard features associated with such factors.

The overall training loss for optimizing network parameters excluding those of the generative adversarial feature disentanglement module is formed by combining equations (8), (9) , and (14):

$$L = L_{\text{dual}} + \alpha L_{\text{int-ce}} + \eta L_{\text{dise}}, \tag{15}$$

where $\alpha$ and $\eta$ are both constant.

## 3.5 Camera-Adaptive Inference Procedure (CAIP)

In the inference phase of our method, we are tasked with identifying the correct matches between a set of gallery images, denoted as $\mathbb{G} = \{\mathbf{G}_i\}_{i=1}^{N_{\text{gal}}}$, and a set of query images, denoted as $\mathbb{Q} = \{\mathbf{Q}_j\}_{j=1}^{N_{\text{que}}}$. We accomplish this by calculating the distances between all pairs of gallery and query images. We feed $\mathbf{G}_i$ and $\mathbf{Q}_j$ into the feature extractor, generating feature vectors $\mathbf{g}_i$ and $\mathbf{q}_j$, respectively. Subsequently, we utilize the camera classifier, trained as part of the interference factor disentanglement module, to assign camera numbers denoted as $c_i^{\text{gal}}$ and $c_j^{\text{que}}$ to $\mathbf{G}_i$ and $\mathbf{Q}_j$, respectively. Considering images captured by the same camera usually share relatively higher visual similarities than those captured by different cameras, we use these predicted camera labels to adjust the cosine distance between $\mathbf{g}_i$ and $\mathbf{q}_j$. The formulation for calculating this distance denoted as $d_{ij}$ is as follows:

$$d_{ij} = \text{dist}(\mathbf{g}_i, \mathbf{q}_j) + \delta \times \mathbb{1}(c_i^{\text{gal}} = c_j^{\text{que}}). \tag{16}$$

Here, we decrease the cosine distance between $\mathbf{g}_i$ and $\mathbf{q}_j$ by a constant threshold denoted as $\delta$ if the inferred camera numbers $c_i^{\text{gal}}$ and $c_j^{\text{que}}$ are consistent. With help of the above distance measurement, we can easily find the gallery images which are most similar to each query image, hence accomplishing the pedestrian re-identification task.

# 4 EXPERIMENTS

## 4.1 Datasets and Metrics

We adopt the following three commonly used CC-ReID datasets (*i.e.*, PRCC, VC-Clothes and LTCC) in our experiments. **PRCC** [39] dataset consists of 33,689 images from 221 identities. Each person in Camera A and B wears the same clothes. While for Camera C, the persons wear different clothes, and the images are taken at different times. **VC-Clothes** [34] is a synthetic dataset, containing 19,060 images rendered by the GTA5 game engine. These images are captured from 512 pedestrian identities under 4 cameras. Each person has 1 to 3 outfits. **LTCC** [30] is an indoor CC-ReID dataset, which has 17,138 images of 152 identities, with 478 different outfits captured from 12 camera views.

We adopt the standard rank-1 accuracy and mean average precision (mAP) for evaluation. In line with prior studies[9, 40], we verify our model through the following three settings: 1) **General Setting**: The gallery set includes both cloth-changing and cloth-consistent samples; 2) **Cloth-Changing Setting**: The gallery set only consists of cloth-changing samples; 3) **Cloth-Consistent Setting**: The gallery set only consists of cloth-consistent samples; For the LTCC and VC-Clothes datasets, we report Rank-1 accuracy and mAP within the general and cloth-changing settings. Regarding the PRCC dataset, we report the Rank-1 accuracy and mAP under the cloth-consistent and cloth-changing settings.

## 4.2 Implementation Details

We adopt ResNet50 [14] pre-trained on ImageNet [5] to form our backbone network. The traditional image parsing method SCHP [24] is utilized to implement the cloth erasing, where we directly perform inference on the training set with the provided pre-trained model. The input images are first resized to $384 \times 192$, and then subjected to random horizontal flipping, random cropping, and random erasing [43]. The training batch size is 32, with 4 person identities and 8 instances for each. The Adam [22] optimizer is used with initial learning rate of $3.5 \times 10^{-4}$ , which is divided by 10 every 20 epochs, for a total of 80 epochs. The discriminator is trained for 2 epochs, and the generator for 5 epochs, with training proceeding in an alternating fashion. The triplet margin $\epsilon$ in Eq. (2) and (6) is set to 0.3, the parameter for momentum update $\gamma$ in Eq. (3), and (4) is set to 0.01, and the temperature factor $\tau$ in Eq. (7) is set to 0.05 across all datasets. In Eq. (15), the parameters $\alpha$ and $\beta$ are set as 1.0, 0.05 for the LTCC dataset, 0.1 and 1.0 for PRCC and VC-Clothes, respectively. The constant threshold $\delta$ in Eq. (16) is set to 0.3 for the LTCC dataset, and 0.05 for both the PRCC and VC-Clothes datasets.

## 4.3 Comparison with State-of-the-art Methods

In the experiments, we compare our approach with existing state-of-the-art (SOTA) methods on the LTCC, PRCC and VC-Clothes datasets. Here, we roughly divide these methods based on whether some auxiliary label information (*i.e.*, Clothes ID, Camera ID) is used or not. Specifically, as shown in Table 1, we compare our method against four methods that use auxiliary labels (*i.e.* CAL [9], AIM [40], CCFA [12], CVSL [29]) and ten methods that do not use auxiliary labels. (*i.e.* HACNN [25], PCB [32], IANet [17], ISP [45], RCSANet [18], FSAM [16], 3DSL [2], GI-ReID [21], AFL [27],

Table 1: Comparison with other state-of-the-art methods on PRCC, VC-Clothes and LTCC datasets.

| Methods | Auxiliary label | PRCC | | | | VC-Clothes | | | | LTCC | | | |
| | | Cloth-Changing | | Cloth-Consistent | | Cloth-Changing | | General | | Cloth-Changing | | General | |
| | | Rank-1 | mAP | Rank-1 | mAP | Rank-1 | mAP | Rank-1 | mAP | Rank-1 | mAP | Rank-1 | mAP |
| HACNN (CVPR18) [25] | None | 21.8 | 23.2 | 82.5 | 84.8 | 49.6 | 50.1 | 68.6 | 69.7 | 21.6 | 9.3 | 60.2 | 26.7 |
| PCB (ECCV18) [32] | None | 41.8 | 38.7 | 99.8 | 97.0 | 87.7 | 74.6 | 62.0 | 62.2 | 23.5 | 10.0 | 65.1 | 30.6 |
| IANet (CVPR19) [17] | None | 46.3 | 45.9 | 99.4 | 98.3 | - | - | - | - | 25.0 | 12.6 | 63.7 | 31.0 |
| ISP (ECCV20) [45] | None | 36.6 | - | 92.8 | - | 72.0 | 72.1 | 94.5 | 94.7 | 27.8 | 11.9 | 66.3 | 29.6 |
| RCSANet (ICCV21) [18] | None | 50.2 | 48.6 | 100 | 97.2 | - | - | - | - | - | - | | |
| FSAM (CVPR21) [16] | None | - | - | - | - | 78.6 | 78.9 | 94.7 | 94.8 | 38.5 | 16.2 | 73.2 | 35.4 |
| 3DSL (CVPR21) [2] | None | 51.3 | - | - | - | - | - | - | - | 31.2 | 14.8 | - | - |
| GI-ReID (CVPR22) [21] | None | - | - | - | - | 64.5 | 57.8 | - | - | 23.7 | 10.4 | 63.2 | 29.4 |
| CAL (CVPR22) [9] | Clothes ID | 55.2 | 55.8 | 100 | 99.8 | 81.4 | 81.7 | 92.9 | 87.2 | 40.1 | 18.0 | 74.2 | 40.8 |
| AFL (TMM'23) [27] | None | 57.4 | 56.5 | 100 | 99.8 | 82.5 | 83.0 | 93.9 | 88.3 | 42.1 | 18.4 | 74.4 | 39.1 |
| AIM (CVPR23) [40] | Clothes ID | 57.9 | 58.3 | 100 | 99.9 | - | - | - | - | 40.6 | 19.1 | 76.3 | 41.1 |
| CCFA (CVPR23) [12] | Clothes ID | 61.2 | 58.4 | 99.6 | 98.7 | - | - | - | - | 45.3 | 22.1 | 75.8 | 42.5 |
| IGCL(TPMI24) [7] | None | 57.8 | 57.4 | - | - | - | - | 82.0 | 83.6 | - | - | 62.0 | 35.0 |
| CVSL(WACV24) [29] | Clothes ID+Viewlabel | 57.5 | 56.9 | 99.1 | 97.5 | - | - | - | - | 44.5 | 21.3 | 76.4 | 41.9 |
| Ours(without Auxiliary label) | None | 67.0 | 62.9 | 99.9 | 98.7 | 92.0 | 88.6 | 93.9 | 91.2 | 40.6 | 18.8 | 74.8 | 39.3 |
| Ours(without Camera ID) | Clothes ID | 70.4 | 66.7 | 100 | 99.3 | 92.2 | 88.8 | 94.4 | 92.0 | 42.9 | 20.7 | 75.9 | 42.6 |
| Ours(without CAIP) | Clothes ID+Camera ID | 71.0 | 66.6 | 100 | 99.1 | 92.2 | 88.9 | 95.3 | 92.0 | 43.6 | 21.0 | 75.1 | 42.4 |
| Ours | Clothes ID+Camera ID | **71.3** | **67.0** | 100 | 99.5 | **94.3** | **90.2** | **96.8** | 93.6 | **50.5** | **25.1** | **80.9** | **46.2** |

IGCL [7]). To reveal the overall performances with/without using auxiliary labels in detail, we have implemented the following four different variants of our method:

**Variant 1** (denoted as without Auxiliary Label): We completely abandon the Generative Adversarial Interference Factor Decoupling Network module proposed in Section 3.4 and the Camera-Adaptive Inference Procedure introduced in Section 3.5.

**Variant 2** (denoted as without Camera ID): We remove the use of Camera ID from the method described in Section 3.4 and discard the Camera-Adaptive Inference Procedure mentioned in Section 3.5.

**Variant 3** (denoted as without CAIP): We only remove the Camera-Adaptive Inference Procedure used for inference, as described in Section 3.5.

**Variant 4**: represents the full assembly of our method.

The overall experimental results are reported in Table 1. We can clearly see that our proposed method surpasses almost all the methods, which outperforms the second-best method by a large margin of 4.74% Rank-1 accuracy and 3.55% mAP on average, over the three datasets under six experimental settings. Compared to other variants of our method without using auxiliary labels, we can conclude that, the designed strategy which properly utilizes the auxiliary cloth and camera IDs, helps a lot to improve the recognition performances. Even without using these auxiliary labels, our method can also outperform existing state-of-the-art method, by a margin of 3.0% Rank-1 accuracy and 1.2% mAP under the same evaluation condition.

## 4.4 Ablation Study

**Effectiveness of the Three Novel Components**. There are three novel components in our proposed method, *i.e.*, 1) the discriminative cloth-irrelevant identity feature learning component (DSIFLF), which optimize the backbone model using the proposed prototype-based cross-stream matching loss $L_{mm}$ and the adaptive cloth-irrelevant contrastive learning loss $L_{cicl}$; 2) the GAIFDN module

Table 2: Ablation study of individual component on PRCC, VC-Clothes and LTCC under cloth-changing setting.

| | Components | | | PRCC | | VC-Clothes | | LTCC | |
| Index | Baseline | DSIFLF | GAIFDN | CAIP | Rank-1 | mAP | Rank-1 | mAP | Rank-1 | mAP |
| 1 | ✓ | - | - | - | 59.0 | 57.7 | 90.0 | 85.7 | 38.3 | 16.5 |
| 2 | ✓ | ✓ | - | - | 67.0 | 62.9 | 92.0 | 88.6 | 40.6 | 18.8 |
| 3 | ✓ | - | ✓ | - | 60.4 | 61.2 | 91.2 | 86.5 | 38.6 | 20.3 |
| 4 | ✓ | ✓ | ✓ | - | 71.0 | 66.6 | 92.2 | 88.9 | 43.6 | 21.0 |
| 5 | ✓ | ✓ | ✓ | ✓ | 71.3 | 67.0 | 94.3 | 90.2 | 50.5 | 25.1 |

Table 3: Ablation study of different loss terms in Eq. (8) of DSIELF on LTCC and PRCC under cloth-changing setting.

| | Components | | | | PRCC | | LTCC | |
| Index | Baseline | $L^*_{cicl}$ | sg($\omega_i$) | $L_{mm}$ | Rank-1 | mAP | Rank-1 | mAP |
| 1 | ✓ | - | - | - | 59.0 | 57.7 | 38.3 | 16.5 |
| 2 | ✓ | ✓ | - | - | 63.4 | 61.0 | 39.0 | 18.0 |
| 3 | ✓ | ✓ | ✓ | - | 64.1 | 61.2 | 40.1 | 18.8 |
| 4 | ✓ | ✓ | ✓ | ✓ | 67.0 | 62.9 | 40.6 | 18.6 |

for discriminative feature enhancement; 3) the camera-adaptive inference module CAIP. To reveal how each component contributes the performance improvement, we have conducted extensive ablation study of individual component on all the three datasets under cloth-changing setting. Specifically, the Baseline method in Table 2 means that we just train this dual-stream backbone network with the traditional cross-entropy loss $L_{ce}$ in Eq. (1) and triplet loss $L_{tri}$ in Eq. (2). Experimental results shown in Table 2 greatly demonstrate the effectiveness of each component in our proposed method.

**Effectiveness of Various Loss Terms in DSIFLF**: To validate the role of various components in DSIFLF, we evaluated the impact of the **adaptive cloth-irrelevant contrastive learning loss** without the weighting coefficient $L^*_{cicl}$, the results with the added weighting coefficient sg($\omega_i$), and the effects of the prototype-based

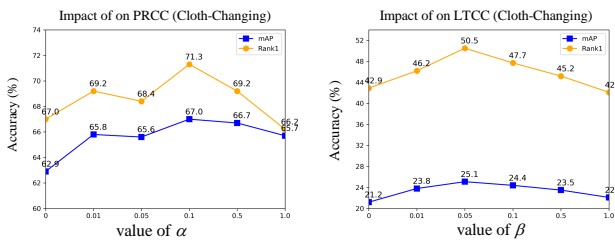

Figure 3: Parameter sensitivity analysis on $\alpha$ in Eq. (15) and $\beta$ in Eq. (8), on PRCC and LTCC under cloth-changing setting.

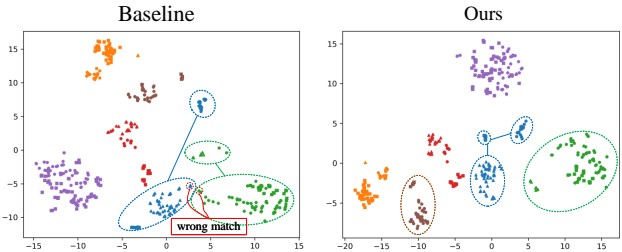

Figure 4: t-SNE visualization for six randomly selected identities is presented, where different colors represent different pedestrians, and within each color, different shapes indicate different clothing items.

cross-stream matching $L_{\text{mm}}$ in Table 3. It can be observed that our $L_{\text{cicl}}^*$ improved the rank1/mAP by (0.7%/1.5%, 4.4%/3.3%) on both datasets. By introducing the weighting coefficient $\text{sg}(\omega_i)$, our model shows an increase by (0.9%/0.7%, 0.7%/0.2%), which indicates that the incorporation of $\text{sg}(\omega_i)$ can alleviate the issues of model training precision degradation due to noise factor from human parsing images. Furthermore, with the introduction of $L_{\text{mm}}$, our model achieves an improvement relative to Index 3 by 2.9%/1.7% in PRCC, demonstrating that our $L_{\text{cicl}}$ has made substantial gains by the proposed prototype-based cross-stream matching loss $L_{\text{mm}}$.

## 4.5 More Discussions

**Parameter Sensitivity Analysis.** We analyze the sensitivity of two key parameters in our method, *i.e.*, the weight $\alpha$ in Eq. (15) and $\beta$ in Eq. (8). We tune the values of each parameter while keeping the others fixed on the LTCC and PRCC datasets. The results are shown in Fig. 3. The results indicate that if the model excessively focuses on the extraction of irrelevant features, or pays too much attention to the matching between pedestrian images and clothes-erased images, then the precision of the model will decrease. Based on these experiments, we set the parameters $(\alpha, \beta)$ to $(1.0, 0.1)$ for the LTCC dataset, and to $(0.05, 1.0)$ for PRCC and VC-Clothes, respectively.

**T-SNE Visualization.** We visualize the t-SNE map of six randomly selected identities, as shown in Fig. 4. The t-SNE map provides a comparative analysis between Ours and the Baseline methods. As shown in the baseline method, some different identities have features that are very close to each other, while the same

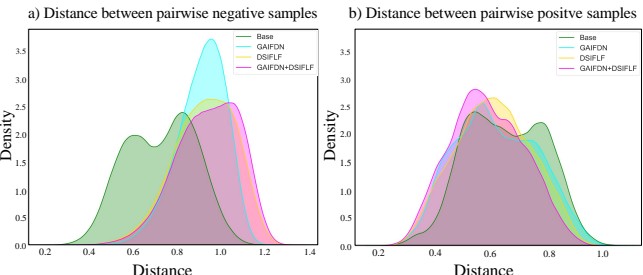

Figure 5: The distribution of $200,000$ **negative sample pairs (a) and $200,000$ positive sample pairs with cross-camera changes and clothing changes (b) in the PRCC dataset is analyzed. By progressively integrating our DSIFLF and GAIFDN into the training framework, the distribution of negative samples gradually shifts to the right (increasing distance), while the distribution of positive samples gradually shifts to the left (decreasing distance).**

identity with different clothes has features that are farther apart, leading to erroneous matches during the query (as in the left image with the blue and green identities). Our method increases the distance between features of different identities and enhances the compactness of features within the same identity (as in the right image with the brown and blue).

**Visualization of Sample Distance Distribution.** We visualize the cosine distance distribution of randomly selected $200,000$ negative sample pairs (a) and $200,000$ positive sample pairs, as shown in Fig. 5. By incrementally integrating the following modules (i.e., GAIFDN, DSIFLF) into the training framework, the peak of the negative sample distance distribution gradually shifts to the right (indicating distance becomes larger). At the same time, the distance distribution of positive samples gradually shifts to the left (indicating a decrease in distance). By comparing the magnitude of this shift, our proposed method primarily acts on increasing the distance of negative sample pairs. This greatly demonstrate the effectiveness of each component in our framework to reduce the interference effects for CC-ReID.

## 5 CONCLUSION

This paper proposes a dual-stream identity feature learning framework to explore discriminative cloth-irrelevant identity feature representations for CC-ReID. Specifically in the cloth-erasure stream, the cloth-irrelevant features can be learned through the proposed adaptive cloth-irrelevant contrastive learning objective, by capturing compact intra-pedestrian structure relationships to mitigate drastic variances caused by clothes characteristics. Meanwhile, to further enhance the discriminability of the internal-identity features, we are the first to propose to address this issue from a reverse perspective by disentangling interference-factor-related identity features for discriminative learning. Specially, we achieve identity feature disentanglement through the proposed adversarial generative noise decoupling network. In the future, we will extend our disentanglement-based method to other cross-modality ReID tasks, as well as some domain generation problems.

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

Received 20 February 2007; revised 12 March 2009; accepted 5 June 2009