# OpenReview forum: "Disentangling Identity Features from Interference Factors for Cloth-Changing Person Re-identification"
_acmmm.org/ACMMM/2024/Conference — MM2024 Poster_

### Official Review · Reviewer_yQmj · 2024-05-04

**Rating:** 4
**Confidence:** 4

**Summary:**

The paper propose a method for Cloth-Changing Person Re-ID. It aims at disentangling identity-related features from interference factors like clothing and camera.

**Strengths:**

- Writing is clear and easy to follow
- The idea of disentangling ID-related features from clothing or camera is not novel, but the paper presents some novelty in the method of doing so, e.g. using the idea of CycleGAN to generate interference-factor-oriented features then suppress it from global features
- Experimental results are decent, outperforming previous methods by around 4% in R-1 acc.
- Ablation studies on the proposed components are quite adequate.
- Analysis is solid with deep insights into the influence of several details in the framework.

**Limitations:**

- Why using Prototype-based Cross-stream Matching (but not existing techniques, even simple ones like KL divergence) to pull two feature distributions closer is not clarified, making it unconvincing.

- For all cross-entropy losses (eq. 1 and 9), in the equations, no ground truth labels is input, which is not correct. Normally, both the predicted class and the ground truth class are passed into the CE loss, however, in eq.1 for example, only predictions $p_i$ and $\hat{p}_i$ of raw image and cloth_erased image are input.

- How cloth labels are constructed is not clarified. Is the number of cloth labels equal to the total number of cloth variations in the entire training set (as similar to CAL paper)?

- For feature extractor $F_{int}$, I am unconvinced how it can be trained to specifically recognize interference factors. No rationale for this is provided.

- The inference process is not technically correct.

       + First, for Re-ID task, it is required that for the same ID, the camera label of the query must be different from that of gallery images. The proposed distance conflicts with the requirement of the task.
       + Second, the claim "images captured by the same camera usually share relatively higher visual similarities" does not make sense for images of the different ID, making the proposed equation 16 of distance unconvincing.

- The used datasets are quite small-scale. Evaluation on very large-scale datasets with diverse distributions in data like DeepChange [1] would be more convincing. This also validates generalizability of the method.

- Unclear implementation details:

       + Architectures of the generators and discriminators are not provided
       + The authors say: "The discriminator is trained for 2 epochs, and the generator for 5 epochs with training proceeding in an alternating fashion". This is pretty unclear:
                 (1) whether the gens and discs are trained along with the entire framework (if so, in which stage they are trained? For the remaining epochs, they are frozen?) or separately (if so, before or after?)
                 (2) how could it be alternating while the number of epochs is not equal (2 vs 5)?
       + Why set the constant threshold in eq. 16 to 0.3 for LTCC and 0.5 for the rest two datasets? Is that after trials and errors?

[1] Xu, Peng, and Xiatian Zhu. "Deepchange: A long-term person re-identification benchmark with clothes change." ICCV. 2023.

**Suitability:**

3

---

### Official Review · Reviewer_HQtq · 2024-05-09

**Rating:** 2
**Confidence:** 4

**Summary:**

This paper proposes to disentangle identity features from interference factors brought by clothes and camera view changes in cloth-changing person ReID. It adopts a dual-stream identity feature learning framework and introduces a generative adversarial interference factor decoupling network, which can relieve the dependence on interference-factor-related identity information. Experimental results show its effectiveness.

**Strengths:**

-	The paper addresses CC-ReID from an interesting perspective of disentangling interference-factor-related identity features for discriminative learning.
-	Results on three datasets show its effectiveness.

**Limitations:**

-	Many recent SOTA methods [20, a, b, c, d] are not compared in Tab. 1. Some of them achieve much better results without using camera labels or clothing labels than the proposed method, but they are not discussed adequately.

[a] Dual Level Adaptive Weighting for Cloth-Changing Person Re-Identification

[b] Semantic-aware Consistency Network for Cloth-changing Person Re-Identification

[c] Exploring Fine-Grained Representation and Recomposition for Cloth-Changing Person Re-Identification

[d] Exploring Shape Embedding for Cloth-Changing Person Re-Identification via 2D-3D Correspondences

-	Eq. 16 in the paper is inconsistent with the implementation in the code. In Line 77 of “test.py”, “distmat” is multiplied rather than added. There are also other inconsistencies in the code, e.g., Lines 76-77 of “train.py”. Therefore, the results of the paper are doubtful.

-	“We decrease the cosine distance” as mentioned in Line 632, but why it is an addition operation in Eq. 16. Moreover, the testing protocol of ReID would exclude the samples with the same camera labels for each person in the query and gallery sets, so is changing the distance between samples with the same inferred camera numbers and different identities like a trick in testing?

-	Ablation studies of each component in GAIFDN are absent, making it hard to understand the contributions of each design.

-	The value of $\eta$ in Eq. 15 is not discussed.

-	The proposed method uses different values for different hyperparameters ($\alpha$, $\beta$, $\delta$) on different datasets. As shown in the ablation studies, different values have a great impact on performance, which makes the method unstable and hard to tune for the best hyperparameters in application.

-	There are some inconsistent results in the paper. For example, different cloth-changing results of LTCC in Tabs. 1 (Ours(without Auxiliary label)) and 3 (Index 4); different optimal values of parameters between the text description and Fig. 3 in Parameter Sensitivity Analysis.

**Suitability:**

3

---

### Official Review · Reviewer_37Dc · 2024-05-21

**Rating:** 4
**Confidence:** 3

**Summary:**

The paper proposes a novel method for cloth-changing person re-identification. They design a dual-stream identity feature learning framework to explore discriminative and cloth-irrelevant identity feature representations.

**Strengths:**

The motivation is clear. The performance is significantly improved and experiments are abundant.

**Limitations:**

1、	The model is relatively complex. So, the time complexity and parameters should be analyzed.
2、	The authors employ the cloth-erasure stream. How much impact does this operation have? What color will this operation erase the clothes into? Do different colors have different effects?
3、	There are many new studies about cloth-changing person re-identification. So, you should compare several new papers instead of unimportant methods.

**Suitability:**

2

---

### Official Review · Reviewer_tUQa · 2024-05-21

**Rating:** 6
**Confidence:** 4

**Summary:**

This paper proposes a disentangle-based method for Cloth-Changing Person ReID by disentangling identity features from interference factors brought by clothes and camera view changes while ensuring robustness and discriminability. Specifically, It adopts a dual-stream identity feature learning framework consisting of a raw image stream and a cloth-erasing stream, to explore discriminative and cloth-irrelevant identity feature representations. The proposed network targets capturing identity-related information residing in the interference factors and disentangling the identity features from such information.

**Strengths:**

1)	The proposed method is novel and proposes disentangling identity features from interference factors for CC-ReID.
2)	The experimental results of this paper are superior to those of SOTA methods, and the ablation study is sufficient, which carefully illustrates the effectiveness of all the modules in the proposed method.
3)	This paper is well written. Each component of the proposed method is clearly defined and described in detail, making it easy for readers to follow.

**Limitations:**

1)	In the adaptative cloth-irrelevant contrastive loss, as shown in Eq.(6), what is the difference between Eq.(6) and the traditional triplet loss? And why the weighting parameter w in Eq.(6) is not trainable, which defined as “sg(w)”?
2)	What motivates Eq.(14)? Why does it use the difference value as the input for cross-entropy loss?
3)	The training procedure for the Generative adversarial feature disentanglement module is unclear, especially for Eq(11-13).
4)	In Eq.(16), how to set the hyper-parameter \sigma?
5)	In Eq.(15), how to set the hyper-parameter \eta? The parameter sensitivity analysis is required.

**Suitability:**

3

---

### Meta-Review · Area_Chair_PTxB · 2024-06-29

**Recommendation:** Accept (Poster)
**Confidence:** 5

**Metareview:**

The paper initially got accept, two borderline accept and one weak reject. The authors have provided a rebuttal. After checking the rebuttal and comments of the other reviewers, three reviewers are satisfied with the rebuttal and gave two accept and one weak accept. However, the Reviewer HQtq who gave weak reject is still not convinced with the rebuttal due to four reasons. 1) The authors deliberately do not compare with DLAW [a] on the LTCC dataset, which largely outperforms the proposed method. However, in the rebuttal, the authors claimed that the proposed method consistently outperforms DLAW; 2) inconsistent between code and implementation details; 3) missing ablation study; and 4) manipulate the ReID protocol.

The AC has carefully checked the rebuttal and found that (2) and (4) have been solved and (3) has been partially solved. However, the (1) issue is indeed existed: the authors deliberately do not report the results of DLAW on LTCC dataset and claimed that the proposed method consistently achieves higher results than DLAW on all datasets. Indeed: DLAW (Table I)=35.56%mAP>>this paper (Table I) = 25.1%mAP). Taking the above consideration, the AC thinks this paper is not yet ready for publication in MM and should fairly reconsider the comparison with state-of-the-art methods in next version.

***TPC Addendum ***
The paper received a split in recommendations. Given the importance of the topic, high average scores, and trend toward increase after rebuttal, the TPC suggests that the debate/discussion continue at the conference with a poster presentation.